# Tree-Climbing Behavior of a Forest-Dwelling Ungulate: The Formosan Serow

**DOI:** 10.3390/ani14152159

**Published:** 2024-07-24

**Authors:** Hayato Takada, Nick Ching-Min Sun, Yu-Jen Liang, Jen-Hao Liu, Ching-Kuo Liu, Kurtis Jai-Chyi Pei

**Affiliations:** 1Wildlife Management Center, Tokyo University of Agriculture and Technology, 3-5-8 Saiwaicho, Fuchu 183-8509, Tokyo, Japan; 2Institute of Wildlife Conservation, National Pingtung University of Science and Technology, 1, Shuehfu Road, Neipu 91201, Pingtung, Taiwan; nicksun99@gmail.com (N.C.-M.S.); b10132074@gmail.com (J.-H.L.); 3Taiwan Wildlife Society, 4th Fl., No. 1, Min-Chiang St., Pingtung City 90051, Pingtung, Taiwan; dahsingliang@gmail.com; 4Hsinchu Branch, Forest and Nature Conservation Agency, Ministry of Agriculture, No. 2, Zhong-Shan Rd., Hsinchu City 30046, Taiwan; wulai9030@yahoo.com.tw

**Keywords:** Formosan serow, *Capricornis swinhoei*, tree climbing, forest-dwelling ungulate, social media platform, Taiwan

## Abstract

**Simple Summary:**

Tree climbing is an extremely rare behavior among ungulates and has rarely been reported in forest-dwelling ungulates. We collected 15 tree-climbing records of Formosan serows, mainly from social media platforms. They climbed trees to forage throughout the year in many locations in Taiwan. This is the first report of tree-climbing behavior in the Formosan serow, which is typically a forest dweller.

**Abstract:**

Ungulates are terrestrial herbivores, basically adapted to running fast on the ground; tree-climbing behavior has been reported only in seven species, and five of them live in open habitats (*Capra hircus*, *C. aegagrus*, *C. falconeri*, *C. cylindricornis*, *Oreotragus oreotragus*). Tree-climbing behavior may also be evolved in ungulates inhabiting dense forests with abundant trees; however, this has rarely been reported in such species (*Moschus leucogaster*, *M. moschiferus*), probably due to the difficulty of observing in the wild. The numerous publicly available records in social networks hold potentially valuable information on the atypical behaviors of wild ungulates. Here, we explored the tree-climbing behavior of a forest-dwelling ungulate, the Formosan serow in Taiwan, a subtropical island, by extracting information from online social media platforms. We researched images and videos of Formosan serows through Facebook and YouTube and collected a total of 15 tree-climbing events. In these materials, Formosan serows climbed 10 tree species, including evergreen coniferous and broad-leaved trees, and a variety of parts, ranging in height from 0.6 to 4 m, and from branches of shrubs to trunks of tall trees. Tree-climbing behavior was recorded throughout Taiwan and from lowlands to subalpine zones, suggesting that tree climbing may be a common behavior in this species. Foraging while climbing trees was frequently observed (53.3%), suggesting that the purpose or benefit for climbing is to obtain additional food other than plants growing near the ground surface. In contrast to other tree-climbing ungulates, Formosan serows climbed trees not only in winter, but also in other seasons, when food is relatively abundant. This is the first scientific report of tree-climbing behavior in the Formosan serow that is typically a forest dweller.

## 1. Introduction

Tree-climbing behavior is widespread in many mammals, and arboreal mammals have various morphological and behavioral adaptations to avoid falling from trees [1,2]. Ungulates are terrestrial herbivores that live in various environments of the world including forests, grasslands, and deserts [3]. Their morphs are adapted to running fast on land, not for climbing trees [4,5]. Therefore, tree-climbing behavior is extremely rare in this group and has been reported only in seven species: feral goat (*Capra hircus*: [6]), wild goat (*C. aegagrus*: [7]), markhor (*C. falconeri*: [8,9,10]), East Caucasian tur (*C. cylindricornis*: [11]), klipspringer (*Oreotragus oreotragus*: [12]), white-bellied musk deer (*Moschus leucogaster* [13]), and Siberian musk deer (*M. moschiferus* [14]). All these species have in common that they are adapted to broken terrains such as steep slopes, cliffs, and rocky areas [10,14,15]. Since five of these species mainly live in open lands, observing their behavior is easier than in forest-dwelling ungulates. On the other hand, tree-climbing behavior may also have evolved in ungulates inhabiting dense forests with abundant trees; however, tree-climbing behavior has not been reported in such forest dwellers species except for two species of musk deer [13,14]. To understand the life history and behavioral adaption of this atypical behavior in ungulates, it is necessary to collect information on more species.

In recent years, the number of videos and photos of wildlife behaviors in the wild uploaded to online platforms and social network platforms has increased dramatically [16,17]. This abundant publicly available information contains potentially valuable data on unusual behaviors in ungulates that are difficult to observe [18], including forest-dwelling species. Therefore, the aim of this study is to describe the tree-climbing behavior of a forest-dwelling ungulate, the Formosan serow (*Capricornis swinhoei*), by extracting information from online platforms.

The Formosan serow is a monomorphic small-sized ungulate (body weight: 21 kg, [19]) that belongs to the subfamily Caprinae. They inhabit all kinds of forests from tropical, subtropical, warm temperate, to subalpine forests in Taiwan [20,21,22]. This species has thick and short legs and soft hooves adapted to broken terrains and prefers habitats with steep slopes and rocky cliffs [20,21]. However, their behavioral aspects are poorly studied due to the low visibility of their habitats and low density, and to date, the tree-climbing behavior of the Formosan serow has not been described in the literature, despite having been known to field biologists in Taiwan for some years. 

## 2. Materials and Methods

We performed research on images and videos of Formosan serows’ tree-climbing behaviors through Facebook (https://www.facebook.com, accessed on 1 July 2024) and YouTube (https://www.youtube.com/, accessed on 1 July 2024), both in Chinese and English. Search terms included “Formosan serow”, “Taiwanese serow”, “Capricornis swinhoei”, “台灣長鬃山羊”, “台灣野山羊”, and “長鬃山羊”. When we detected target photos or videos, we collected as much as possible of the following information: the date and time of day (daytime or evening), location and the altitude of the image, and the stratification (tree layer or shrub layer) and DBH (cm) of the tree, parts (branch or trunk) and minimum diameter (cm) of the trees on which the Formosan serows stood, the maximum climb height (m) from the base of the tree, and the behavioral pattern on the trees. We also attempted to identify the climbed trees’ species or genus level. For the serows in the images, age class was estimated from horn length, and sex was identified from external genitalia [23] whenever possible. 

Behavioral patterns were categorized into foraging, alerting (the animal lifts its head above the body axis, looking intently at and orienting the ears toward the photographer), flight (running away from the photographer), and resting (ruminating or resting while standing). When foraging on the tree was confirmed, we tried to identify the foraging items. When flight behavior was confirmed, we recorded the height (m) from the point of jump to the ground. In addition to social media, one of the authors (JHL) photographed the tree-climbing behavior of Formosan serows in two cases, so these were also included in this report.

We assumed that an adult Formosan serow of either sex has the following morphometric values: the length of the head and body is approximately 100 cm; the shoulder height is approximately 60 cm; the tail length is approximately 7 cm; and the horn length is approximately 9 cm [24]. All the values of length or height (cm, m) were estimated from the sizes of the Formosan serows in the video or image. All the values of length or height (cm, m) taken from the video or image were estimated by comparing them with the size of Formosan serows using Adobe Photoshop Element 2018 (16.0). If the length or height could not be measured accurately, the minimum length that could be determined from the video or image was recorded. 

## 3. Results

A total of 15 images of tree-climbing behaviors of Formosan serows were collected from online platforms (13 cases) and direct observation by the author (2 cases) (Table 1, Figure 1). Ten species of trees were identified, including evergreen coniferous trees and evergreen broad-leaved trees, and the stratification included both tree layers (33.3%) and shrub layers (66.7%). The mean DBH ± SD of the climbed tree was 21.2 ± 21.6 cm (range: 5–60 cm). Tree-climbing behavior was confirmed in a wide range of locations from southern (Pingtung County and Chiayi County) to eastern (Taitung County), central (Taichung City), and northern (Hsinchu City and Miaoli Country) Taiwan, and at elevations from 200 m to 3500 m. The tree-climbing behavior was exhibited both in the daytime (76.9%) and evening (23.1%) throughout the year (Jan., Feb., Jun., Aug., Sep., and Dec.).

All the tree-climbing behaviors recorded were of only one individual at a time, and all were adults. In only two cases, we were able to determine the gender, one male and one female, of the serow. They could climb both on branches (80.0%) and trunks (20.0%), and the mean minimum diameter ± SD of the trees on which the Formosan serows stood was 15.4 ± 23.1 cm (range: 2–60 cm). When they climbed on a branch, they did so by clamping the branch between their main hooves (Figure 1). The mean maximum climbing height ± SD was 1.9 ± 0.9 m (range: 0.6–4 m). Foraging behavior was the most frequently observed (53.3%), followed by alerting (26.7%), flight (13.3%), and resting (6.7%). For the eight cases in which foraging was confirmed, seven were feeding on leaves (five cases) and lichen (two cases) on the climbed trees, and one was feeding on leaves of adjacent trees. In two cases of flight behavior, they jumped to the ground from heights of 1.5 m and 2.5 m, respectively. 

## 4. Discussion

Based on the images we collected, Formosan serows climb various trees, including evergreen coniferous trees and evergreen broad-leaved trees, and a variety of parts from branches of shrubs to trunks of tall trees, ranging in height from 0.6 to 4 m. Such tree-climbing behavior was observed in a wide range of regions and altitude zones (lowlands to subalpine) in Taiwan, suggesting that tree climbing is a common behavior in this species. All the tree species identified in this study are common in the elevation zones where the images were taken [25,26,27,28,29]. This is the first scientific report on tree-climbing behavior in the Formosan serow, which is typically a forest-dwelling ungulate.

The Formosan serow prefers broken terrains [21], which is consistent with seven ungulate species that have been reported to exhibit tree-climbing behavior [6,7,10,11,12,13,14]. Some ungulates adapted to steep terrain have flexible hooves and muscular short legs that are suitable for moving on unstable grounds [10,30,31]. They also have retinas that are good at vertical object recognition [32]. These traits adapted to steep terrain seem to enable tree climbing, as well. 

There are three other species of the genus *Capricornis* (*C. crispus*, *C. rubidus*, and *C. sumatraensis*) that are also forest-dwelling and prefer broken terrain; however, details of their behaviors have been understudied. Although one of the authors (HT) has conducted direct observation of the Japanese serow, which is closely related to the Formosan serow [33], for over 12 years [34,35,36,37,38,39,40], tree-climbing behavior in Japanese serows has never observed or heard of. Therefore, there may be variations in the occurrence of tree-climbing behavior among these closely related species, and it might have evolved in the Formosan serow independently as an adaptation to the local environments in Taiwan. The body weight of the Formosan serow is 21 kg, which is the lightest compared with the mainland serow (85–140 kg), Japanese serow (31–48 kg), and the red serow (110–160 kg) [24]; therefore the light body weight of the Formosan serow might be an advantage to develop such behaviors so to expand their activity into a three-dimensional space. 

In addition, foraging on trees was frequently observed in Formosan serows, suggesting that climbing is likely to obtain additional food other than plants growing near the ground surface. Foraging behavior on trees has been reported in all tree-climbing ungulates [6,7,10,11,12,13,14]. It has been reported that tree-climbing behavior by the markhor [8,10] and klipspringer [12] occurs during the winter or dry seasons when terrestrial food resources are scarce. Interestingly, we confirmed tree-climbing behavior in Formosan serows not only in winter, but also in spring, summer, and autumn, when food is relatively abundant. This commonly used foraging technique in Formosan serows, therefore, might also allow them to obtain food items of better quality, or even allow them to browse. In the future, it will be necessary to examine the relationship between the occurrence of tree-climbing behavior, food habits, and forage quality with the season and habitat characteristics. However, it cannot be ruled out that they may climb trees for purposes other than foraging, such as avoiding terrestrial predators or simply exploring.

The present study contains several limitations. The sample size was small, at 15, and data was biased towards locations near human activity due to social media use. In the future, direct observation or biologging, such as GPS collars with a video and/or accelerometer, will be needed in the field to assess when, where, who, and how wild Formosan serows climb trees.

## 5. Conclusions

We found tree-climbing behavior in a forest-dwelling ungulate, the Formosan serow, which is an extremely rare behavior among ungulates. Our results suggest that they may climb trees to obtain additional forage throughout the year in various environments, from tropical to subalpine forests, of Taiwan.

## Figures and Tables

**Figure 1 animals-14-02159-f001:**
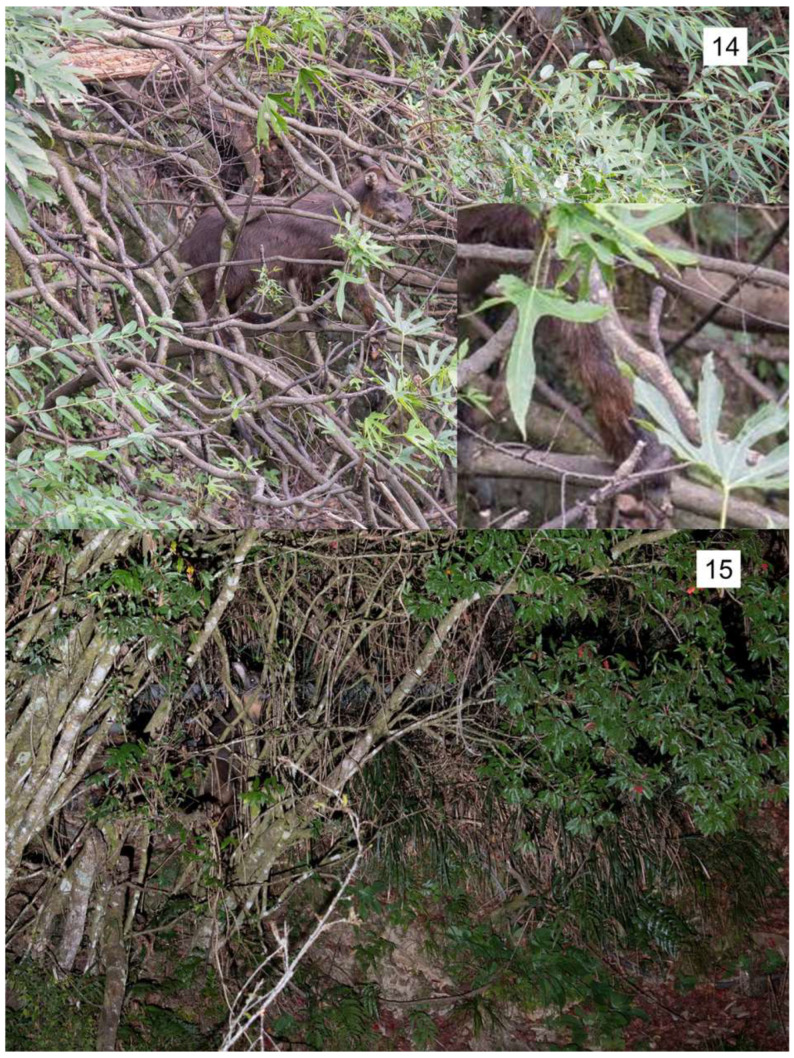
Tree-climbing Formosan serows (*Capricornis swinhoei*). Numbers in the images indicate the ID of the image.

**Table 1 animals-14-02159-t001:** Characteristics of 15 cases of tree-climbing behavior in the Formosan serow (Capricornis swinhoei) extracted from online platforms and observation by the authors.

ID	Trees Climbed by Serow					Location Name	Altitude
	Species		Type	Stratification	DBH (cm)		
1	*Deutzia pulchra*		EB	Shrub layer	5	unk.	unk.
2	*Symplocos modesta*		EB	Tree layer	60	unk.	unk.
3	unk.		EB	Tree layer	60	Hsinchu City, Northern Taiwan	unk.
4	unk.		EB	Tree layer	60	Mountain Beidawu, Pingtong County, Southern Taiwan	1500–3000
5	*Prinsepia scandens*		EB	Shrub layer	5	unk.	unk.
6	*Salix fulvopubescens*		EB	Tree layer	5	unk.	unk.
7	*Salix fulvopubescens*		EB	Shrub layer	5	unk.	unk.
8	*Abies kawakamii*		EC	Tree layer	30	Yushan National Park, Chiayi County, Southern Taiwan	2944
9	*Oreocnide pedunculata*		EB	Shrub layer	10	unk.	unk.
10	*Mallotus japonicus*		EB	Shrub layer	10	Changbin Township, Taitung County, Eastern Taiwan (Coastal Mountain Range)	<200
11	*Mallotus japonicus*		EB	Shrub layer	10	Changbin Township, Taitung County, Eastern Taiwan (Coastal Mountain Range)	<200
12	*Tsuga chinensis* var. *formosana*		EC	Shrub layer	7.5	Yushan National Park, Chiayi County, Southern Taiwan	2944
13	*Abies kawakamii*		EC	Shrub layer	30	Xue Mountain, Miaoli Country, Northern Taiwan	3500
14	*Fatsia polycarpa*		EB	Shrub layer	10	Wuling Veterans Farm, Taichung City, Central Taiwan	2200
15	*Photinia niitakayamensis*		EB	Shrub layer	10	Daxue Mountain, Taichung City, Central Taiwan	2300
ID	Date	Time of Day	Climbing Individual		Serow Location in the Tree		
			Age Class	Sex	Parts	Minimum Diameter (cm)	Height (m)
1	unk.	daytime	Adult	M	Branch	2	0.6
2	2023/6/18	daytime	Adult	unk.	Trunk	60	2.4
3	unk.	daytime	Adult	unk.	Trunk	60	4
4	2019/2/16	daytime	Adult	unk.	Trunk	60	?
5	August	daytime	Adult	unk.	Branch	4	3
6	August	daytime	Adult	unk.	Branch	4	1.8
7	September	daytime	Adult	unk.	Branch	4	1.8
8	2022/12/6	daytime	Adult	unk.	Branch	5	1.8
9	unk.	evening	Adult	F	Branch	5	1.2
10	2022/8/4	daytime	Adult	unk.	Branch	5	1.8
11	2022/8/10	evening	Adult	unk.	Branch	5	1.8
12	2023/2/23	evening	Adult	unk.	Branch	4	1.8
13	unk.	daytime	Adult	unk.	Branch	4	0.6
14	2021/9/4	daytime	Adult	unk.	Branch	4	1.8
15	2019/1/30	evening	Adult	unk.	Branch	5	1.8
ID	Behavioral Pattern		Food Item		Image Type	Social Network	
1	AL, FO, FL		*Deutzia pulchra* (climbed tree)		Video	Facebook	
2	AL, FL		NA		Video	YouTube	
3	AL		NA		Video	YouTube	
4	RE		NA		Video	YouTube	
5	FO		*Prinsepia scandens* (climbed tree)		Video	YouTube	
6	FO		*Salix fulvopubescens* (climbed tree)		Video	YouTube	
7	AL		NA		Video	YouTube	
8	FO		Lichen, moss, or usnea (on the branch of climbed tree)		Video	Facebook	
9	NA		NA		Photo	Facebook	
10	FO		*Mallotus japonicus* (climbed tree)		Photo	Facebook	
11	FO		*Mallotus japonicus* (climbed tree)		Photo	Facebook	
12	FO		Lichen, moss, or usnea (on the branch of climbed tree)		Video	Facebook	
13	FO		*Juniperus squamata* Lamb. (not climbed tree)		Photo/Video	Facebook	
14	NA		NA		Photo	NA	
15	NA		NA		Photo	NA	

EB = evergreen broad-leaved tree, EC = evergreen coniferous tree, M = male, F = female, AL = alerting, FO = foraging, FL = flight, RE = resting, unk. = Unknown, NA = not applicable. Data sources, 1: https://www.facebook.com/story.php?story_fbid=6452372754857377&id=100002541491686 (accessed on 1 July 2024), 2: https://www.youtube.com/watch?v=h9WxAtg5_70 (accessed on 1 July 2024), 3: https://www.youtube.com/watch?v=itxpi5WISfY&list=FL-Bah4XdS7i1v0cXBSv6nuw&index=3&t=12s (accessed on 1 July 2024), 4: https://www.youtube.com/watch?v=HSjyZpHutik&list=FL-Bah4XdS7i1v0cXBSv6nuw&index=5 (accessed on 1 July 2024), 5, 6, and 7: https://www.youtube.com/watch?v=lPHex-v8E7c&list=FL-Bah4XdS7i1v0cXBSv6nuw&index=1 (accessed on 1 July 2024), 8: https://www.facebook.com/meihsiuhwang/videos/1649020045604154 (accessed on 1 July 2024), 9: https://www.facebook.com/groups/388953801293069/permalink/1037867366401706 (accessed on 1 July 2024), 10 and 11: https://www.facebook.com/JuhuFarmstay/posts/pfbid02jTYowMDDrenrDFh33X3dGMvxd4h5ZxR3Wq93oB21S5a3n8Vg4EYFBZBTiw8foW33l (accessed on 1 July 2024), 12: https://www.facebook.com/meihsiuhwang/videos/632577902180234 (accessed on 1 July 2024), 13: https://www.facebook.com/photo/?fbid=10160021339187659&set=pcb.10160021340402659 (accessed on 1 July 2024), 14 and 15: Photographed by authors.

## Data Availability

Data generated or analyzed during this study are included in this published article (Appendix A).

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
