# Peer review of "Tree-Climbing Behavior of a Forest-Dwelling Ungulate: The Formosan Serow"

_animals, 2024, doi:10.3390/ani14152159_

Round 1
Reviewer 1 Report
Comments and Suggestions for Authors
Specific remarks
L. 16-17 – “Tree-climbing is extremely rare behavior among ungulates, and has not been reported in forest dwelling ungulates” – This statement is not right. There is another forest-dwelling ungulate, Musk deer, which commonly climb on trees. This behavior of climbing trees is published many times in different papers. Please, open Google Scholar and you will find several such papers regarding musk deer.
L. 19-20 – “This is the first report of tree-climbing behavior in a forest dwelling ungulate.” – This is wrong statement.
L. 21-22 – “…, and tree climbing behavior has been reported only in five species that live in open habitats.” – Maybe it would be better to indicate which species they are.
L. 37-38 – “This is the first scientific report of tree-climbing behavior in a forest dwelling ungulate.” – the same, this is not first scientific report for the forest-dwelling ungulate species. Please, see many papers devoted to musk deer.
L. 48-50 – “it has been reported only in five species; feral goat (Capra hircus: [6]), wild 48 goat (C. aegagrus: [7]), markhor (C. falconeri: [8-10]), east Caucasian tur (C. cylindricornis: 49 [11]), and klipspringer (Oreotragus oreotragus: [12]). – The tree climbing is usual for Capra nubiana (e.g. “Coimbra, J.P., Alagaili, A.N., Bennett, N.C., Mohammed, O.B. and Manger, P.R., 2019. Unusual topographic specializations of retinal ganglion cell density and spatial resolution in a cliff‐dwelling artiodactyl, the Nubian ibex (Capra nubiana). Journal of Comparative Neurology, 527(17), pp.2813-2825.).
L. 177-178 – “The present study contains several limitations. The sample size was as small as 15, 177 and data was biased towards locations near the human activity due to social media use” – This is absolutely right statement. It would be good to investigate this tree climbing behavior of the Taiwan serow and use field observation data, not only published photo and video from media.
Comments on the Quality of English Language.
Author Response
Reviewer 1
Specific remarks
- 16-17 – “Tree-climbing is extremely rare behavior among ungulates, and has not been reported in forest dwelling ungulates” – This statement is not right. There is another forest-dwelling ungulate, Musk deer, which commonly climb on trees. This behavior of climbing trees is published many times in different papers. Please, open Google Scholar and you will find several such papers regarding musk deer.
↓
Thank you for your important comment. We have collected information about tree climbing behavior of musk deer (Moschus leucogaster, M. moschiferus), added that information, and revised throughout the manuscript.
Changes: L18, 21-22, 24-25, 27, 41, 50, 52-53, 55, 58-59, 167, 204-205, 246-249
- 19-20 – “This is the first report of tree-climbing behavior in a forest dwelling ungulate.” – This is wrong statement.
↓
We have revised as “This is the first report of tree-climbing behavior in the Formosan serow that is typical forest dweller”.
Changes: L21-22
- 21-22 – “…, and tree climbing behavior has been reported only in five species that live in open habitats.” – Maybe it would be better to indicate which species they are.
↓
I have revised this statement and added scientific name of each species.
Changes: L25, 27
- 37-38 – “This is the first scientific report of tree-climbing behavior in a forest dwelling ungulate.” – the same, this is not first scientific report for the forest-dwelling ungulate species. Please, see many papers devoted to musk deer.
↓
As mentioned above, we have revised as “This is the first report of tree-climbing behavior in the Formosan serow that is typical forest dwellers”.
Changes: L41
- 48-50 – “it has been reported only in five species; feral goat (Capra hircus: [6]), wild 48 goat (C. aegagrus: [7]), markhor (C. falconeri: [8-10]), east Caucasian tur (C. cylindricornis: 49 [11]), and klipspringer (Oreotragus oreotragus: [12]). – The tree climbing is usual for Capra nubiana (e.g. “Coimbra, J.P., Alagaili, A.N., Bennett, N.C., Mohammed, O.B. and Manger, P.R., 2019. Unusual topographic specializations of retinal ganglion cell density and spatial resolution in a cliff‐dwelling artiodactyl, the Nubian ibex (Capra nubiana). Journal of Comparative Neurology, 527(17), pp.2813-2825.).
↓
We never mentioned that Capra nubiana climbs trees.
- 177-178 – “The present study contains several limitations. The sample size was as small as 15, 177 and data was biased towards locations near the human activity due to social media use” – This is absolutely right statement. It would be good to investigate this tree climbing behavior of the Taiwan serow and use field observation data, not only published photo and video from media.
↓
We have agreed with your comment and have stated as “In the future, direct observation or biologging, such as GPS collars with video and accel-erometer will be needed in the fields to assess when, where, who, and how wild Formosan serows climb trees.”.
Changes: L200-202
Reviewer 2 Report
Comments and Suggestions for Authors
Manuscript ID: animals-3122597
Title: Tree climbing behavior of a forest dwelling ungulate, Formosan serow
Authors: Hayato Takada *, Nick Ching-Min Sun, Yu-Jen Liang, Jen-Hao Liu, Ching-Kuo Liu, and Kurtis Jai-Chyi Pei *
Review
Aim of this study was is to describe the tree climbing behavior of a forest dwelling ungulates, Formosan serow (Capricornis swinhoei). For this, authors extracted information from online platforms.
Tree climbing in ungulates is not fully investigated, therefore manuscript has merit, scientific priority and is relevant to investigations of ungulate ecology. There are very few references on the subject, so authors present also references related to other ungulate species. We located one more possible reference, however it is not available for open access, therefore authors might try it:
Chen YiMing, C. Y., He WunLing, H. W., Wong RueiHong, W. R., & Kuo ChaoNien, K. C. (2015). Mammal assemblages of different forest types in the Cilan cypress forest. Taiwan Journal of Forest Science, 2015, Vol. 30, No. 1, 75-88 ref. 39
There is no specific approach oh hypotheses, as authors collected material from the internet. This method is documented, links for most of the sources are given as Supplemental table S1.
Figures and tables are appropriate to the text.
Conclusion summarizes observations of the climbing behavior of the studies species, Formosan serow (Capricornis swinhoei).
As material was collected from internet, ethical statement is not required.
General comments
1. Perhaps is possible to remove Table S1 and incorporate link to sources into Table 1. If there are problems to fit into page width, links can be given in footnotes.
2. Figure 1 is very interesting, perhaps it can be given page width.
Rating of the manuscript
Novelty: great, no former publications on the studied issues were found
Scope: narrow and targeted, fits to Animals section Wildlife
Significance: high; results do not need interpretation, as there are no comparisons available with similar studies
Quality: presented according rules of scientific publishing
Interest to readers: high, but is limited to limited number of investigators
Overall merit: high
English level: acceptable and understandable (I am not native speaker)
Other comments:
1. Line 126: Figure caption must be not in bold.
2. Line 129: Latin name in italics.
3. Line 130: plat homes?
Comments on the Quality of English Language
minor mistakes or mistypes
Author Response
Reviewer 2
Title: Tree climbing behavior of a forest dwelling ungulate, Formosan serow
Authors: Hayato Takada *, Nick Ching-Min Sun, Yu-Jen Liang, Jen-Hao Liu, Ching-Kuo Liu, and Kurtis Jai-Chyi Pei *
Review
Aim of this study was is to describe the tree climbing behavior of a forest dwelling ungulates, Formosan serow (Capricornis swinhoei). For this, authors extracted information from online platforms.
Tree climbing in ungulates is not fully investigated, therefore manuscript has merit, scientific priority and is relevant to investigations of ungulate ecology.
There are very few references on the subject, so authors present also references related to other ungulate species. We located one more possible reference, however it is not available for open access, therefore authors might try it:
Chen YiMing, C. Y., He WunLing, H. W., Wong RueiHong, W. R., & Kuo ChaoNien, K. C. (2015). Mammal assemblages of different forest types in the Cilan cypress forest. Taiwan Journal of Forest Science, 2015, Vol. 30, No. 1, 75-88 ref. 39
↓
Thanks for the reference. We have added this citation.
Changes: L70, 264-265
There is no specific approach oh hypotheses, as authors collected material from the internet. This method is documented, links for most of the sources are given as Supplemental table S1.
Figures and tables are appropriate to the text.
Conclusion summarizes observations of the climbing behavior of the studies species, Formosan serow (Capricornis swinhoei).
As material was collected from internet, ethical statement is not required.
General comments
- Perhaps is possible to remove Table S1 and incorporate link to sources into Table 1. If there are problems to fit into page width, links can be given in footnotes.
↓
We have removed Table S1 and added this information into Table1 as foot notes.
Changes: Table1, L137-154
- Figure 1 is very interesting, perhaps it can be given page width.
↓
We have expanded to fit page width.
Changes: Figure 1
Rating of the manuscript
Novelty: great, no former publications on the studied issues were found
Scope: narrow and targeted, fits to Animals section Wildlife
Significance: high; results do not need interpretation, as there are no comparisons available with similar studies
Quality: presented according rules of scientific publishing
Interest to readers: high, but is limited to limited number of investigators
Overall merit: high
English level: acceptable and understandable (I am not native speaker)
Other comments:
- Line 126: Figure caption must be not in bold.
↓
We have revised as your suggestion.
Changes: L131-132
- Line 129: Latin name in italics.
↓
We have revised as your suggestion.
Changes; L134
- Line 130: plat homes?
↓
We have revised as “platforms”.
Changes: L134